# Effects of Enzamin, a Microbial Product, on Alterations of Intestinal Microbiota Induced by a High-Fat Diet

**DOI:** 10.3390/nu14224743

**Published:** 2022-11-10

**Authors:** Toshinori Yasuzawa, Ryota Nishi, Satono Ishitani, Osamu Matsuo, Shigeru Ueshima

**Affiliations:** 1Department of Nutrition, School of Human Cultures, The University of Shiga Prefecture, Hikone 522-8533, Japan; 2Department of Food Science and Nutrition, Faculty of Agriculture, Kindai University, Nara 631-8505, Japan; 3Faculty of Medicine, Kindai University, Osaka-Sayama 589-9511, Japan

**Keywords:** intestinal microbiota, adhesion molecule, inflammation, Enzamin

## Abstract

In the human intestinal tract, there are more than 100 trillion microorganisms classified into at least 1000 different species. The intestinal microbiota contributes to the regulation of systemic physiologic functions and the maintenance of homeostasis of the host. It has been reported that the alteration of the intestinal microbiota is involved in metabolic syndromes, including type II diabetes and dyslipidemia, inflammatory bowel disease, allergic disease, and cancer growth. It has been reported that a microbial product from *Paenibacillus polymyxa* AK, which was named Enzamin, ameliorated adipose inflammation with impaired adipocytokine expression and insulin resistance in db/db mice. In order to investigate the effect of Enzamin on the intestinal microbiota and inflammation induced by obesity, mice were fed with a high-fat diet and 1% Enzamin for 4 weeks. Enzamin improved the *Firmicutes*-to-*Bacteroidetes* ratio and altered the intestinal microbiota in mice fed the high-fat diet. In addition, Enzamin suppressed the decreased expression of claudin-4 and the increased serum LPS level in mice fed with the high-fat diet. Modulating the intestinal microbiota with Enzamin may cause a decrease in serum LPS level. Based on these results, Enzamin may improve inflammation and metabolic disorders by regulating the intestinal microbiota in obese mice.

## 1. Introduction

In the human intestinal tract, there are more than 100 trillion microorganisms classified into at least 1000 species [1]. More than 99% of intestinal microbiota are anaerobes, such as *Bacteroides*, *Clostridium,* and *Bifidobacterium*. The intestinal microbiota includes useful microorganisms such as *Bifidobacterium* and destructive microorganisms such as *Clostridium*. Based on the balance of these microorganisms, the environment in the human intestinal tract is maintained in a healthy condition. However, since the intestinal microbiota is associated with energy metabolism or the systemic homeostasis of the host human, an imbalance in these microorganisms can cause several systemic disorders. It has been reported that the alteration of the intestinal microbiota is involved in obesity [2,3], metabolic syndromes including type II diabetes [4,5,6], inflammation [7], asthma [8,9], and cancer growth [10].

It is well known that fermented foods and fermented ingredients contain probiotics and/or prebiotics and influence the intestinal microbiota. Probiotics consist of viable bacteria, such as *Lactobacillus*, *Bacillus subtilis,* and *Clostridium butyricum*, and may have healthy effects on the host human by influencing the balance of intestinal microbiota. Prebiotics stimulate the growth of beneficial bacteria in the intestinal tract and modulate the composition and activity of the intestinal microbiota [1]. Two indigestible oligosaccharides (fructooligosaccharides), inulin and oligofructose, are among the prebiotics [11,12]. Therefore, probiotic management of the intestinal microbiota can be supplemented with prebiotics. Both probiotics and prebiotics ameliorate the imbalanced environment of the intestinal microbiota, which results in promoting the health of the host human.

Enzamin is a microbial product from *Paenibacillus polymyxa* AK. *Paenibacillus polymyxa* is a Gram-positive bacterium found in the environment, such as in soil, water, and plant roots [13]. *Paenibacillus polymyxa* produces exopolysaccharides (EPSs). EPSs from *Paenibacillus polymyxa* have potential applications in healthcare fields [14]. EPSs from *Paenibacillus polymyxa* reduced sucrose-induced hyperglycemia [15]. A total of 100 mL of *Paenibacillus polymyxa* AK strain preincubated for 18 h was inoculated to 40 L of culture medium which consisted of 1.5% cornstarch broken down with amylase, 11% sucrose as a carbon source, and 0.3% yeast extract as a nitrogen source. The culture medium was incubated statically at 30 °C for 1 week and subsequently cold incubated at 10 °C for another week. Then, the culture medium was autoclaved at 121 °C for 15 min and used as Enzamin stock solution. Enzamin contains many types of components, such as polysaccharides, oligosaccharides, organic acids, nucleic acids, and peptides. In a previous report, it was demonstrated that Enzamin exhibited profibrinolytic and antithrombotic properties by increasing the release of t-PA from endothelial cells and might prevent thrombotic disorders [16]. Furthermore, Enzamin ameliorated adipose inflammation, with impaired adipocytokine expression and insulin resistance in db/db mice [17]. It was demonstrated that Enzamin improved insulin resistance and might prevent type II diabetes. Thus, Enzamin is thought to be a beneficial supplement for suppressing the development of metabolic syndromes. Although the alteration of the intestinal microbiota is involved in metabolic syndromes, the effects of Enzamin on the intestinal microbiota have not yet been clarified. In the present study, we investigated the beneficial effects of Enzamin on the altered intestinal microbiota induced by obesity in mice.

## 2. Materials and Methods

### 2.1. Reagents

Enzamin liquid (lot no. 170116), a microbial product, was provided by the Enzamin Laboratory Co., Ltd., Osaka, Japan. All other reagents and chemicals were of the highest grade available.

### 2.2. Animal Experiments

All animal experimental protocols were approved by the Kindai University Experimental Animal Committee (approval number KAAG-29-002) and followed the animal experiment guidelines of Kindai University. The mice were freely fed and given water while they were kept in an environment with a 12 h light/dark cycle at 22 ± 2 °C. Five-week-old male C57BL/6J mice (Japan SLC, Inc., Shizuoka, Japan) were divided into 3 groups: (1) control group, receiving normal diet (13% of kcal from fat; CE-2, CLEA Japan Inc., Tokyo, Japan) and tap water; (2) high-fat diet (HFD) group, receiving HFD (60% of kcal from fat; HFD32, CLEA Japan Inc., Tokyo, Japan) and tap water; and (3) HFD ENZ group, receiving HFD32 and tap water containing 1% of Enzamin (Enzamin Research Institute, Osaka, Japan). After 4 weeks of treatments, the stools were collected for the analysis of the microbiota, and large intestine mucosa was collected for the analysis of adhesion molecules.

### 2.3. Isolation of DNA from Stool

The isolation of DNA from stool was performed by using a QIAamp DNA Stool Mini Kit (Qiagen, Hilden, Germany), according to the manufacturer’s instructions.

### 2.4. Sequencing of 16S rRNA Gene and Microbiome Analysis

Two-step PCR was performed on the purified DNA samples to obtain sequence libraries. The first PCR was performed to amplify with primer pairs of Bakt_341F: CCTACGGGNGGCWGCAG and Bakt_805R: GACTACHVGGGTATCTAATCC corresponding to the V3–V4 region of the 16S rRNA gene. The raw data extraction of PCR amplicons by Illumina MiSeq was performed at Macrogen (Kyoto, Japan). The relative abundance of identified taxa in each sample was also analyzed at Macrogen (Kyoto, Japan) [18].

### 2.5. Blood Analyses

After 4 weeks of treatments, blood samples were collected for the analysis of serum lipopolysaccharide (LPS), triglyceride (TG), and total cholesterol (TC). LPS, TG, and TC were measured by using commercial kits, ELISA for LPS (Cloud-Clone Corp., Katy, TX, USA), Wako Triglyceride E-test, and Wako Cholesterol E-test (both from Fujifilm Wako Pure Chemical Corporation, Osaka, Japan).

### 2.6. Immunoblotting

Large intestine mucosal lysates were separated by 10% SDS-polyacrylamide gels and blotted onto polyvinylidene fluoride (PVDF) membranes. After blocking, the membranes were incubated with anti-occludin (Novus Biologicals, Centennial, CO, USA), anti-Claudin-4 (Proteintech, Rosemont, IL, USA), and anti-β-actin (Cell Signaling, Danvers, MA, USA) overnight at 4 °C. The membranes were washed and incubated with horseradish- peroxidase-conjugated secondary antibodies for 1 h at room temperature. The protein-antibody complex was detected using ECL reagent (SuperSignal^TM^ West Dura Extended Duration Substrate, Thermo Fisher Scientific, Waltham, MA, USA), and signals were detected using an LAS 4000 mini imager (Fujifilm, Tokyo, Japan).

### 2.7. Statistics

Statistical analyses between 3 groups were performed using one-way analysis of variance (ANOVA) and post hoc Tukey–Kramer multiple comparisons, and statistical analyses between 2 groups were performed using Student’s *t*-test utilizing Microsoft Excel (Microsoft Corporation, Redmond, WA, USA) and the add-in software Statcel 4 (OMS Publishing Inc., Saitama, Japan). The data are shown as mean ± SD for each group. Significant differences were considered at *p* < 0.05.

## 3. Results

Body weight was significantly increased in mice fed with a high-fat diet compared to control mice fed with a normal diet. Enzamin did not affect the body weight of mice fed with a high-fat diet (Table 1).

The serum LPS levels were significantly higher in mice fed with a high-fat diet than control mice fed with a normal diet. Enzamin significantly decreased the serum LPS levels compared to mice fed with a high-fat diet (Table 2). The serum TG and TC levels in mice fed with a high-fat diet were significantly elevated compared to mice fed with a normal diet. However, the serum TG and TC levels were significantly reduced by Enzamin in mice fed with a high-fat diet (Table 2).

The individual differences in microbial composition were taxonomically evaluated at the phylum level (Figure 1a). The mean of individual relative abundance control was compared between control mice and mice fed with a high-fat diet and Enzamin-treated mice (Figure 1b). At the phylum level, the high-fat diet increased the relative abundance of *Firmicutes* (*p* < 0.05, 25.6 ± 2.9% vs. 36.4 ± 7.9%) and reduced the relative abundance of *Bacteroidetes* (*p* < 0.01, 70.9 ± 3.4% vs. 41.5 ± 6.5%). On the other hand, Enzamin decreased the relative abundance of *Firmicutes* (*p* < 0.05, 36.4 ± 7.9% vs. 25.2 ± 3.0%) and increased the relative abundance of *Bacteroidetes* (*p* < 0.01, 41.5 ± 6.5% vs. 54.6 ± 1.5%) in mice fed with a high-fat diet. Mice fed with a high-fat diet had a significantly increased *Firmicutes*-to-*Bacteroidetes* ratio compared to mice fed with a normal diet. However, mice treated with Enzamin had a significantly reduced *Firmicutes*-to-*Bacteroidetes* ratio compared to mice fed with a high-fat diet (Figure 1c). At the family level, the high-fat diet significantly increased the relative abundance of *Ruminococcuceae*. However, Enzamin significantly decreased the relative abundance of *Ruminococcuceae* in mice fed with a high-fat diet (Figure 2a). At the genus level, *Akkermansia* was not detectable in control mice. Although *Akkermansia* was detected in mice fed with a high-fat diet, by comparison, the concomitant administration of Enzamin with the high-fat diet significantly decreased the relative abundance of *Akkermansia* (Figure 2b).

To investigate the adhesion molecules in intestinal epithelial cells, we measured the expression of claudin-4 and occludin in intestinal mucosa. The expression of claudin-4 was significantly decreased in mice fed with a high-fat diet compared to control mice fed with a normal diet. Enzamin partially normalized claudin-4 expression (Figure 3a). No statistically significant difference in the expression of occludin was observed between the control, HFD, and HFD ENZ groups (Figure 3b).

## 4. Discussion

It is considered that inflammation and metabolic disorders in high-fat-diet-induced obesity are associated with changes in intestinal microbiota. In order to investigate the influence of Enzamin on the intestinal microbiota in high-fat-diet-induced obese mice, the mice were treated with the oral administration of Enzamin. The results show that Enzamin improved the *Firmicutes*-to-*Bacteroidetes* ratio and altered the intestinal microbiota in mice fed with a high-fat diet (Figure 1). The *Firmicutes*-to-*Bacteroidetes* ratio is known to be associated with obesity. Increased abundances of *Firmicutes* and decreased abundances of *Bacteroidetes* were observed in high-fat-diet-induced obese mice and ob/ob mice [19,20]. Moreover, the *Firmicutes*-to-*Bacteroidetes* ratio may be closely associated with the intestinal tight junction barrier [21]. The disruption of the intestinal epithelial barrier causes LPS, which is produced by Gram-negative bacteria, to migrate from the intestinal tract to the bloodstream, which in turn causes systemic inflammation [22]. However, Enzamin downregulated the high *Firmicutes*-to-*Bacteroidetes* ratio induced by the high-fat diet (Figure 1c) and lowered serum LPS levels (Table 2). In addition, although the high-fat diet induced a decrease in the expression of claudin-4, which is an adhesion molecule in tight junctions of intestinal epithelial cells, the concomitant administration of Enzamin increased its expression to almost the control level (Figure 3a). Based on these findings, it was thought that Enzamin improved the *Firmicutes*-to-*Bacteroidetes* ratio and recovered the disruption of the intestinal epithelial barrier induced by the high-fat diet. Therefore, it is suggested that Enzamin may suppress the migration of LPS from the intestinal tract to the bloodstream and reduce serum LPS levels.

In our results, the high-fat diet increased the relative abundance of *Ruminococcuceae* at the family level, while the additional administration of Enzamin with the high-fat diet suppressed the augmentation of *Ruminococcuceae* that was induced by the high-fat diet (Figure 2a). *Ruminococcus gnavus* (*Ruminococcuceae*), which produces inflammatory polysaccharides, is considered to be associated with inflammatory bowel diseases, such as Crohn’s disease. It is reported that polysaccharides produced by *Ruminococcus gnavus* induce the secretion of inflammatory cytokine TNFα through Toll-like receptor 4. *Ruminococcus gnavus* can also utilize mucin, which is a glycoprotein that plays a role in the intestinal barrier [23]. Thus, *Ruminococcuceae* is thought to be a harmful bacterium for humans. Since Enzamin suppresses the augmentation of *Ruminococcuceae* induced by a high-fat diet in mice, it might be a beneficial supplement for humans.

In this study, it was shown that Enzamin reduced *Akkermansia* (at the genus level) in the high-fat condition (Figure 2b). *Akkermansia muciniphila* is well known to be a mucin-degrading bacterium. It was reported that the excessive degradation of mucin by *Akkermansia muciniphila* exacerbated intestinal inflammation [24]. In addition, increased *Akkermansia muciniphila* has been observed in patients with type 2 diabetes [4]. The mucus layer containing mucin is a physical barrier against bacteria and pathogens. The excessive degradation of mucin impairs the barrier function of the intestinal tract and leads to various disorders. Mucin deficiency induces inflammation and has been implicated in the pathogenesis of inflammatory bowel disease [25]. Thus, it is suggested that the growth of *Akkermansia* by a high-fat diet induces dysfunction of the gut barrier and subsequently causes a systemic inflammatory state. However, it may be assumed that Enzamin has the ability to protect against systemic inflammation by suppressing the growth of *Akkermansia*.

Intestinal dysbiosis causes increased blood LPS levels, and the level of LPS binding protein showed positive correlations with body mass index, HbA_1c_, and inflammatory cytokines [26]. It is considered that intestinal dysbiosis induced by a high-fat diet leads to insulin resistance through an LPS-dependent mechanism [27]. It was reported that a high-fat diet increased the translocation of Gram-negative bacteria through the intestinal mucosa to blood and adipose tissue and induced an increase in TNFα, IL-1β, PAI-1, IL-6, and IFN-γ in adipose tissue [28]. Probiotic treatment ameliorated adipose tissue inflammation and insulin resistance induced by a high-fat diet [28]. The expression of TNFα in the adipose tissue of db/db mice was increased by fivefold compared to control mice, but Enzamin treatment significantly decreased the TNFα expression [17]. Enzamin treatment also significantly decreased the expression of MCP-1 and IL-6 in the adipose tissue of db/db mice [17]. Histological analysis showed increased macrophage accumulation in the adipose tissue of db/db mice, and Enzamin significantly suppressed macrophage accumulation [17]. It was also reported that LPS from intestinal microbiota increased macrophage accumulation in adipose tissue and inflammatory markers such as TNFα [29]. Therefore, Enzamin may contribute to reducing macrophage accumulation in adipose tissue and inflammation by lowering blood LPS levels.

High-fat-diet-induced obesity increases tissue inflammatory cytokines and serum TNFα and IL-6 [30]. The limitation of this study is that the direct effects of Enzamin on adipose tissue inflammation were not assessed. In the previous report [17], although serum TNF-α in db/db mice was higher than that in control mice, Enzamin was reported to suppress adipose tissue inflammation and decrease serum TNFα in db/db mice. Moreover, it has also been reported that the expressions of TNF-α mRNA and IL-6 mRNA in epididymal white adipose tissue in db/db mice were higher than those in control mice. However, Enzamin decreased the expressions of TNF-α mRNA and IL-6 mRNA in epididymal white adipose tissue in db/db mice. Therefore, it is speculated that serum inflammatory cytokines are increased by a high-fat diet condition and increased serum inflammatory cytokines may be decreased by Enzamin administration.

Enzamin ameliorated serum TC and TG levels (Table 2), which is consistent with previous reports in db/db mice. Moreover, it has been reported that Enzamin increased the expression of muscle CPT 1b (carnitine palmitoyltransferase 1b) mRNA in db/db mice [17]. Since muscle CPT 1b is involved in lipid oxidation in muscle, Enzamin may have potential for improving lipid profiles. However, the precise mechanisms by which Enzamin improves serum TC and TG levels are not clear. Further investigations are required to clarify the mechanisms.

In the present study, it was clear that the oral administration of Enzamin improved intestinal dysbiosis induced by a high-fat-diet and showed beneficial effects. Although it is possible that Enzamin may be affected and altered by digestive juice or digestive enzymes, it is unknown if Enzamin is altered to what kind of materials at the gut. Further investigations are needed to determine what kind of materials derived from Enzamin alter intestinal microbiota.

In conclusion, the present study suggests that the alteration of intestinal microbiota induced by the oral administration of Enzamin may be one of its advantages in ameliorating adipose tissue inflammation and metabolic disorder.

## Figures and Tables

**Figure 1 nutrients-14-04743-f001:**
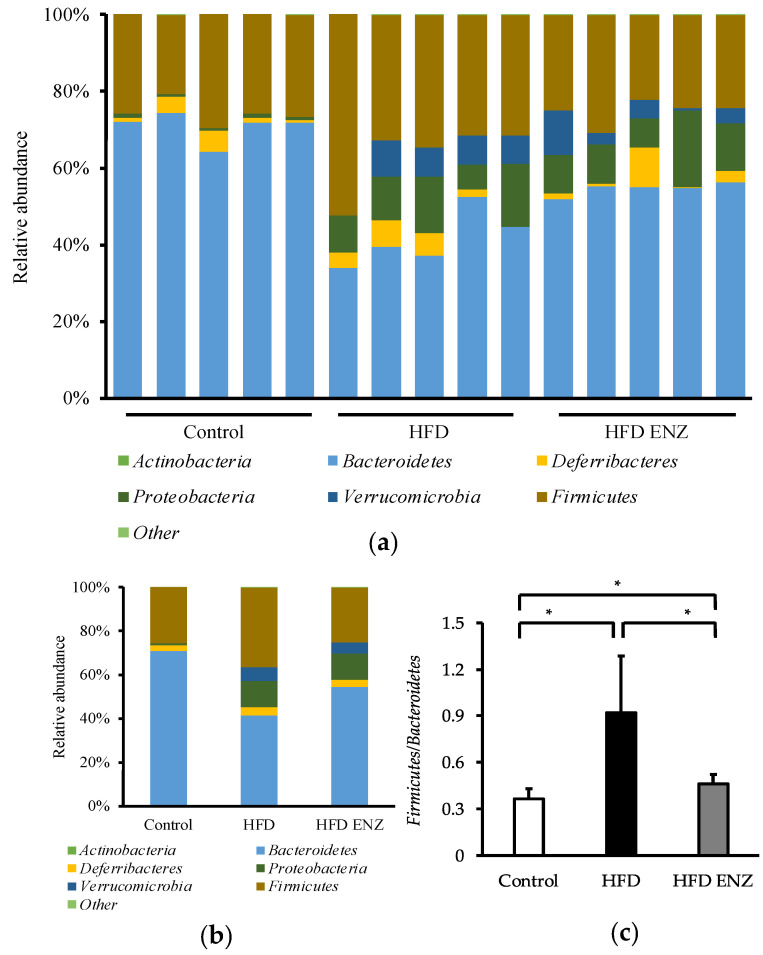
Composition of intestinal microbiota at phylum level. Each component of cumulative bar chart indicates a phylum level. (**a**) Individual composition, (**b**) comparative analysis of composition, and (**c**) *Firmicutes*-to-*Bacteroidetes* ratio. In (**c**), data are shown as mean ± SD, *n* = 5, * *p* < 0.05.

**Figure 2 nutrients-14-04743-f002:**
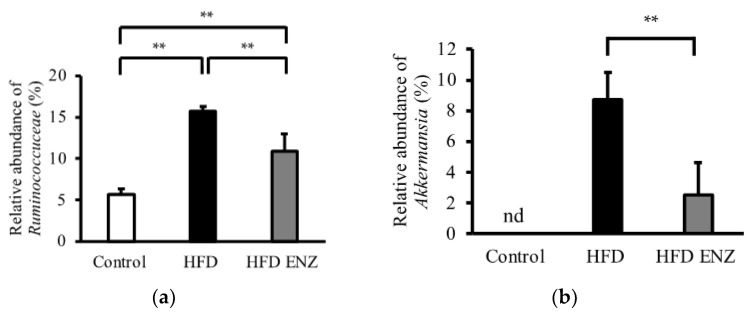
Relative abundance of *Ruminococcuceae* (family level) and *Akkermansia* (genus level). (**a**) Relative abundance of part of *Ruminococcuceae* family was compared between control mice (control), mice fed with high-fat diet (HFD), and Enzamin-treated mice fed with high-fat diet (HFD ENZ). (**b**) Relative abundance of part of *Akkermansia* genus was compared between control, HFD, and HFD ENZ. nd, not detectable. Data are shown as mean ± SD, *n* = 5, ** *p* < 0.01.

**Figure 3 nutrients-14-04743-f003:**
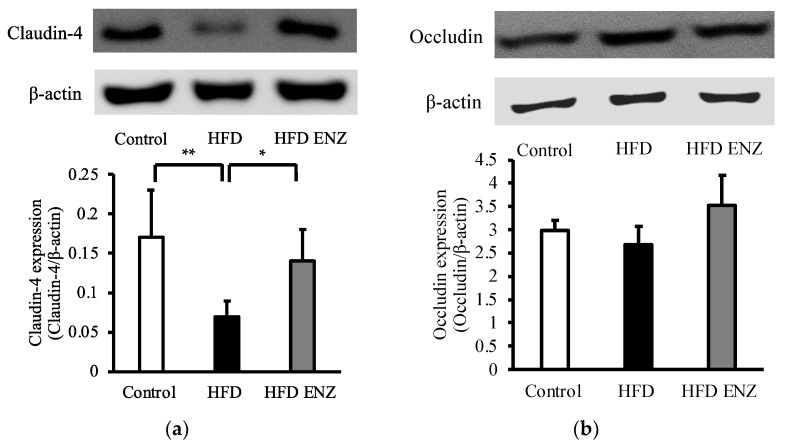
Expression of adhesion molecules in intestinal mucosa. Expression of (**a**) claudin-4 and (**b**) occludin in intestinal mucosa as analyzed by immunoblotting and compared between control mice (control), mice fed with high-fat diet (HFD), and Enzamin-treated mice fed with high-fat diet (HFD ENZ). Data are shown as mean ± SD, *n* = 6, * *p* < 0.05, ** *p* < 0.01.

**Table 1 nutrients-14-04743-t001:** Body weight in experimental period.

	Control	HFD	HFD ENZ
Initial body weight (g)	14.6 ± 1.1	14.5 ± 0.9	14.3 ± 1.0
Body weight after 4 weeks (g)	23.6 ± 1.1	30.0 ± 2.4 **	29.9 ± 3.6 **

Data are shown as mean ± SD, *n* = 10, ** *p* < 0.01 versus control.

**Table 2 nutrients-14-04743-t002:** Blood analysis.

	Control	HFD	HFD ENZ
LPS (ng/mL)	22.3 ± 3.6	63.3 ± 12.4 **	46.0 ± 6.6 **^,#^
TG (mg/dL)	65.9 ± 9.3	125.0 ± 8.4 **	95.1 ± 9.4 **^,##^
TC (mg/dL)	79.0 ± 7.3	131.5 ± 6.8 **	108.4 ± 12.9 **^,##^

Data are shown as mean ± SD, *n* = 4–5, ** *p* < 0.01 versus control; ^#^
*p* < 0.05, ^##^
*p* < 0.01 versus HFD.

## Data Availability

The data presented in this study are available on request from the corresponding author.

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
