# Peer review of "Effects of Enzamin, a Microbial Product, on Alterations of Intestinal Microbiota Induced by a High-Fat Diet"

_nutrients, 2022, doi:10.3390/nu14224743_

Round 1
Reviewer 1 Report
General comment:
1. Alterations in intestinal microbiota have been and continue to be of great interest to basic researchers and clinicians and impact overall health in numerous ways. The current study examines the impact on “Enzamin”, which is a microbial product of Paenibacillus polymyxa, on intestinal microbiota in high-fat diet-fed mice. Major findings include that Enzamin treatment modified the microbiota composition; reduced circulating LPS, triglyceride, and total cholesterol levels; and ameliorated the decrease in intestinal permeability (as indicated by intestinal mucosal claudin-4 protein expression) in high-fat diet-fed mice. The findings are of moderate interest and would be of more significant interest if they were expanded to include immune responses and measures of systemic inflammation.
Comments:
General/language/format
2. There are some awkward language constructions throughout the manuscript, including the misuse or non-use of articles such as “the”, phrasing of “as” that should be “such as”, the use of an adjective as a noun, use of plural nouns when a singular would be more appropriate, subject-verb disagreement, and other errors. The paper needs a very thorough proofread by a qualified editor. I have not commented individually on each error.
3. Please use single spaces after periods consistently throughout the manuscript. In many instances, there appear to be two spaces.
Title
4. The inclusion of “microbial product” is vague. I suggest stating “Enzamin, a microbial product”.
Abstract
No comments
Introduction
5. The introduction gives sufficient background on probiotics and prebiotics and their effect on a healthy microbiota. However, there is a lack of transition to the discussion of Enzamin. Is Paenibacillus polymyxa a commonly occurring bacterial species? If so, has its potential role been elucidated? How was Enzamin extracted from P. polymyxa as a bioactively useful compound? This information will better support the current study.
Methods
6. Line 79: typo. DAN should be DNA.
7. Statistics: Please clarify which data were analyzed using one-way ANOVA and which were analyzed using Student’s t-test. Were these tests determined a priori?
Results
No comments
Discussion
8. Enzamin was administered orally. Is it known if Enzamin is altered in the gut in any way?
9. The discussion describes how Enzamin may affect intestinal bacterial species and blood LPS level. How might Enzamin affect triglyceride or cholesterol levels (both of which are hallmark characteristics of metabolic syndrome)?
10. The final is unsubstantiated by the findings. The study did not directly assess whether Enzamin ameliorated adipose tissue inflammation or metabolic disorder.
Reviewer 2 Report
This is an interesting manuscript in which the authors describe the effect of Enzamine on inflammation and metabolic disorders, focusing on the role of the microbiota. However, I have some suggestions for the authors.
The authors should add some more information on the effect this compound has on counteracting metabolic disorders.
I also strongly recommend a screening of the serum levels of certain inflammatory cytokines (e.g. TNF-alpha, IL-6, IFN) to further support the inflammatory condition that is induced by the hyperlipidic diet.
In my opinion, the manuscript is interesting. It fits perfectly into the field of the journal. I advise the authors to increase the bibliography by adding some more recent manuscripts.
